# Reinforcement Learning Benchmarks for Traffic Signal Control

**James Ault**
Texas A&M University
College Station, TX
jault@tamu.edu

**Guni Sharon**
Texas A&M University
College Station, TX
guni@tamu.edu

## Abstract

We propose a toolkit for developing and comparing reinforcement learning (RL)-based traffic signal controllers. The toolkit includes implementation of state-of-the-art deep-RL algorithms for signal control along with benchmark control problems that are based on realistic traffic scenarios. Importantly, the toolkit allows a first-of-its-kind comparison between state-of-the-art RL-based signal controllers while providing benchmarks for future comparisons. Consequently, we compare and report the relative performance of current RL algorithms. The experimental results suggest that previous algorithms are not robust to varying sensing assumptions and non-stylized intersection layouts. When more realistic signal layouts and advanced sensing capabilities are considered, a distributed deep Q-learning approach is shown to outperform previously reported state-of-the-art algorithms in many cases.

## 1 Introduction

Travel time studies in urban areas show that 12–55% of commute travel time is due to delays induced by signalized intersections (stopped or approach delay) [14, 24]. Hence, optimized signal control has the potential of reducing commute time, traffic congestion, emissions, and fuel consumption, while requiring minimal infrastructure changes.

A signalized intersection is composed of incoming and outgoing roads where each road is affiliated with one or more lanes. A signal controller must assign right of passage to phases, where each phase corresponds to a specific traffic movement through the intersection (incoming to outgoing roads/lanes). Two phases are defined to be in *conflict* if they cannot be enabled simultaneously (their affiliated traffic movement is intersecting). Each intersection serves vehicles which are assumed to continuously arrive on incoming roads. Each vehicle is associated with a specific, outgoing target road. At each time step, the signal controller is tasked with assigning right-of-passage (green signal) to a set of non-conflicting phases such that some utility measurement is optimized. The utility to be optimized is commonly defined as the sum of vehicles' delay imposed by the intersection [20, 3].

Given that signalized intersections vary with regards to their layout and demand profile, optimized control policies may differ and are instance dependent. Consequently, signal controllers usually require to be optimized based on the observed state of the environment. Such online optimization of signal controllers requires: (a) sensing the state of approaching traffic (e.g., number and position of approaching vehicles, approaching speeds, queue length, accumulated delay) aggregated by approaching roads/lanes, and (b) defining a control policy that takes the current state of traffic as input and outputs the next phases to be enabled (which is translated to a green, yellow, and red assignment for each light box).

A line of publications has attempted to harness deep reinforcement-learning (RL) techniques towards this control optimization problem. While several approaches claim state-of-the-art performance [7,

---

18], they fail to compare performance between themselves using a standard testbed. Consequently, it is challenging to determine which algorithm results in state-of-the-art performance and under what circumstances. This paper attempts to address this gap by establishing:

1. An RL testbed environment for traffic signal control that is based on the well-established Simulation of Urban Mobility traffic simulator (SUMO) [5].

2. Benchmark single- and multiagent-signal control tasks which are based on realistic traffic scenarios from SUMO.

3. An OpenAI GYM interface [6] which allows easy deployment of standard RL algorithms.

4. A standardized implementation of state-of-the-art RL-based signal control algorithms.

The presented testbed and benchmark environment denoted *REinforced Signal COntrol* or RESCO is used to provide a first-of-its-kind comparison between state-of-the-art RL-based signal control algorithms. The reported comparative study is performed using the aforementioned, SUMO simulator and scenarios which are inspired by real-world cities and calibrated demands. The reported results paint a picture where algorithms that claim state-of-the-art performance struggle in realistic traffic scenarios and are often outperformed by a decentralized deep Q-learning approach [19].

## 2 Background

Recent publications [15, 25] proposed to utilize state-of-the-art reinforcement learning (RL) algorithms for online optimization of signal controllers. In this approach, the state of the intersection is usually defined by the set of incoming vehicles (incoming lane, speed, waiting time, queue position) and the current signal (right-of-passage) assignment. An RL agent is tasked with optimizing a policy which maps states to signal assignment. Such an approach showed a potential reduction of up to 73% in vehicle delays when compared to fixed-time actuation [20].

### 2.1 Reinforcement learning

In reinforcement learning (RL) an agent is assumed to learn through interactions with the environment. The environment is commonly modeled as a Markov decision process (MDP) which is defined by: $\mathcal{S}$ – the state space, $\mathcal{A}$ – the action space, $\mathcal{P}(s_t, a, s_{t+1})$ – the transition function of the form $\mathcal{P} : \mathcal{S} \times \mathcal{A} \times \mathcal{S} \rightarrow [0, 1]$, $R(s, a)$ – the reward function of the form $R : \mathcal{S} \times \mathcal{A} \rightarrow \mathbb{R}$, and $\gamma$ – the discount factor. The agent is assumed to follow an internal policy $\pi$ which maps states to actions, i.e., $\pi : \mathcal{S} \rightarrow \mathcal{A}$. The agent's chosen action ($a_t$) at the current state ($s_t$) affects the environment such that a new state emerges ($s_{t+1}$) as well as some reward ($r_t$) representing the immediate utility gained from performing action $a_t$ at state $s_t$, given by $R(s, a)$. The observed reward is used to tune the policy such that the expected sum of discounted reward, $J_\pi = \sum_t \gamma^t r_t$, is maximized. The policy $argmax_\pi[J_\pi]$ is the optimal policy and is denoted by $\pi^*$.

Common approaches for training a policy using RL include, value-based, policy-gradient, and actor-critic approaches. A value-based approach attempts to learn the expected future utility from states (*state value*) or from action-state pairs (*action value* or *q-value*). The control policy is then directed towards actions/states that maximize the expected utility ($J_\pi$). A prominent example of a value-based approach is the model-free deep Q-learning algorithm [19]. In the policy-gradient approach [30] a policy is defined through a parameterized differential equation, where the parameters are gradually updated, following the policy gradient, towards favorable outcomes (as experienced through the reward function). Using state or action value estimations for defining favorable outcomes for policy-gradient updates is usually referred to as an actor-critic approach. A prominent example of a state-of-the-art actor-critic approach is the proximal-policy optimization (PPO) algorithm [22] which provides some guarantees regarding monotonicity in the policy improvement over training iterations.

### 2.2 Traffic signal control as an MDP

A signalized intersection is composed of incoming and outgoing roads where each road is assembled from one or more lanes. The intersection is assigned a set of phases, $\Phi$. Each phase, $\varphi \in \Phi$, is affiliated with a specific traffic movement through the intersection, as illustrated in Figure 1. Two

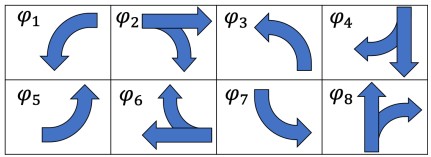

Figure 1: Typical phases, or traffic movements, in a 4-way intersection.

phases are defined to be in *conflict* if they cannot be enabled simultaneously (their affiliated traffic movement is intersecting).

For example, in the phase allocation presented in Figure 1, $\varphi_2$ and $\varphi_1$ are conflicting phases. At each time step, a signal controller is responsible to enable some combination of non-conflicting phases such that a long-term objective function is optimized. When considering RL-based controllers, the signalized intersection environment is commonly modeled through the following MDP.

**State Space** ($\mathcal{S}$): the state space is defined by the state of incoming traffic and the currently enabled phases. Specifically, the state of incoming traffic is defined through the assumed sensing capabilities. These assumptions vary between publications. Some published work [3] assume state-of-the-art traffic sensing technology [13] which allows high-resolution data regarding incoming traffic. For example, real-time observations regarding the number of approaching vehicles, stopped vehicles accumulated waiting time, number of stopped vehicles, and average speed of approaching vehicles. Other publications assume less informative sensing capabilities. For instance by assuming that only the stopped queue length per lane is visible [7], or by assuming that only the waiting time of the first vehicle in the queue is visible [18].

Previous work also varies in the assumed sensing radius. While some assume a sensing radius that covers the entirety of the incoming roads [7],[1] others assume a more realistic 50 meter [18] or 200 meter [3] sensing radius.

**Action Space** ($\mathcal{A}$): at each time-step, the controller chooses a set of non-conflicting phases to be assigned the right-of-passage (green light). If the chosen phases differ from the currently enabled phases then a mandatory yellow phase is enforced for a predefined time duration. Note that assigning yellow phases is not a part of the action space but a constraint imposed on the control sequence by the environment.

**Transition function** ($\mathcal{P}$): the transition function is defined by the traffic progression following the signal assignment. This progression can be defined within a simulated environment following a specific traffic model (as in this paper) or by real-world traffic progression as part of a real-world implementation (out of scope for this paper).

**Reward function** ($\mathcal{R}$): previous publications commonly used (minus) queue length summed over all incoming lanes as their reward function [29]. Such a reward function is simple to implement and is relevant to congestion alleviation. On the other hand, it fails to normalize the benefits from optimized signal operation over travel times. Consequently, other reward functions were suggested. Among them, (minus) total delays imposed by the intersection [23], (minus) waiting time at the intersection [3], and (minus) traffic pressure [18].

### 2.2.1 Multiagent control

In many real-world scenarios traffic flow is sought to be optimized over a road network which includes multiple intersections. In such scenarios, signal control is assumed to be centrally coordinated over all the intersections. These control problems are denoted *multiagent signal control* and are defined by the Cartesian product of the composing intersections' state spaces and action spaces.

Following the curse of dimensionality [4] introduced by a multiagent control task, some researchers take a hierarchical control approach where intersections are partitioned into local control groups [18, 7].

---

[1]Chen et al. 2020 does not explicitly specify the sensing radius. However, the referenced codebase (3/6/2020) uses an unbounded sensing radius.

## 2.3 Related work

Next, we review several RL algorithms that claim state-of-the-art performance for traffic signal control.

**Deep Q-Learning**: deep Q-learning [19] has been proposed for the control of signalized intersections in a number of works [3, 25, 23, 16, 20, 12, 27, 15]. Here we adopt the implementation of [3], which reported up to a 19.4% improvement over an actuated controller. Ault et al. defined a convolutional Q-network where queue length, number of approaching vehicles, total approaching speed, and total waiting time are aggregated in convolutional layers over lanes composing the same incoming road. This implementation uses a (minus) total waiting time reward function.

**MPLight**: MPLight [7] utilizes the concept of pressure to coordinate multiple intersections. Pressure is the difference in queue lengths from incoming lanes of an intersection and the queue length on a downstream intersection's receiving lane. Chen et al. used pressure as both the state and reward for a DQN agent shared over all intersections on top of the FRAP [32] model. The authors reported up to a 19.2% improvement in travel times over the next best compared method, PressLight [28].

**FMA2C**: FMA2C [8] builds on the prior work of MA2C. MA2C enabled cooperation between signal control agents (one per intersection) denoted *workers*. Adjacent workers are coordinated through a local discounted neighborhood reward, discounted neighborhood-appended states, and action fingerprint sharing on otherwise independent advantage actor-critic agents. FMA2C extends this to a hierarchy of *managing agents* on top of the workers. The managing agents are trained to optimize flow within their assigned region. The workers are then trained to incorporate the high-level goals of their managing agent. The authors reported up to a 6% improvement in average delays over the *Greedy controller* described in Section 3.4.

### 2.3.1 Evaluation environments for RL-based signal controllers

For their experimental evaluation, previous publications relied on custom-made scenarios that were, in many cases, tailored for the evaluated RL algorithm. Jinming and Feng 2020 used the well-established simulation of urban mobility (SUMO) environment. SUMO is widely accepted in the transportation community and—as we also note—is a reasonable testbed choice. Jinming and Feng did provide results on a scenario that is based on a real-world city (Monaco). However, the reported scenario was based on a modified Monaco scenario which contains an addition of 18 synthetic traffic signals beyond the official "MoST" scenario [9] and also includes non-validated inflated traffic demands. We extend on that work by introducing benchmark signal control tasks that are based on validated traffic scenarios while providing a flexible interface that is suitable for a variety of RL algorithms. Moreover, our codebase is publicly available allowing researchers to perform meaningful comparative evaluations.

Zhang et al. 2019 presented their own simulation testbed denoted *CityFlow*. This evaluation testbed suffers from two main drawbacks, (1) as opposed to SUMO, CityFlow is not rigorously calibrated and evaluated by the general transportation community. CityFlow is claimed to produce equivalent output as SUMO. However, those claims are based on results from simplified grid network scenarios; (2) a common benchmark scenario in CityFlow is the Manhattan, NY network. This scenario is claimed to represent real-world city layout and demand. However, support for this claim is limited.

Other relevant publications [21] presented evaluations that are based on the autonomous intersection management (AIM) simulator [10]. The main drawback of the AIM simulator is the lack of traffic scenarios which are based on real-world cities. AIM commonly produces a simple grid network with symmetric intersections. Again, one might claim that such a grid network is akin to the road layout in Manhattan, NY, yet a deeper analysis of traffic trends is required to support such claims and their relevancy to the real world.

# 3 The Reinforced Signal Control (RESCO) toolkit

We present a standard RL traffic signal control testbed denoted reinforced signal control or RESCO. The main goals of this standard testbed are:

1. Provide benchmark single- and multiagent-signal control tasks which are based on well-established traffic scenarios.
2. An OpenAI GYM interface [6] within the testbed environment which allows easy deployment of state-of-the-art RL algorithms.
3. A standardized implementation of state-of-the-art RL-based signal control algorithms.

RESCO is open source and free to use/modify under the GNU General Public License 3. The code is built on top of SUMO-RL [1] and is available on Github at `github.com/Pi-Star-Lab/RESCO`. The embedded traffic scenarios are distributed with their own licenses. Cologne based scenarios are under creative commons BY-NC-SA and Ingolstadt based with the GNU General Public License 3.

## 3.1 State and action Space

In order to allow a variety of sensing assumptions (as described in Section 2.2, "State space"), RESCO assumes the most advanced sensing capabilities [3]. The user can pull any subset of the state features based on specific sensing assumptions. Specifically, per state, per intersection, per lane, the system provides a dictionary with the following features: stopped vehicles queue length, number of approaching vehicles (not stopped), total waiting time for stopped vehicles, sum of approaching vehicles' (not stopped) speed, maximum waiting time (over stopped vehicles), number of arrivals (added vehicles during last time step), number of departures (vehicles removed during last time step). On top of that, the user can also define the effective sensing distance on initialization.

For each intersection the provided benchmark control task defines the set of non-conflicting phases. The action space is the set of non-conflicting phase combinations as defined in Section 2.2, "Action space". By default acts are chosen for the next 10 seconds of simulation, with the first 3 seconds reserved for yellow signals if necessary, these values follow Ma and Wu [18]. The interface accepts a unique intersection ID and an index representing the chosen phase combination to be enabled.[2]

## 3.2 Reward metrics

In order to allow maximal flexibility for the user, the interface allows designating any of the reward metrics defined in Section 2.2 "Reward function", as well as a custom weighted combination of these metrics. When initializing a control task, the user can pass a weight vector as an argument where each entry designates the weight of one of the metrics in the reward function. The weight vector entries are defined as follows. 1: system travel time, 2: approximated signal induced delays,[3] 3: total waiting time at intersections, 4: average queue length, 5: traffic pressure.

## 3.3 Benchmark control tasks

The presented signal control benchmark tasks are based on two well-established SUMO scenarios, namely, "TAPAS Cologne" [26] and "InTAS" [17]. Both scenarios describe traffic within a real-world city, Cologne and Ingolstadt (Germany) respectively, including road network layout and calibrated demands. The road network for these scenarios is shown in Figure 2. These scenarios were chosen as they are well-accepted by the transportation community and include a congested downtown zone with multiple signalized intersections. Note that RESCO can easily be fitted to other SUMO scenarios.

Three benchmark control tasks are considered per traffic scenario, namely, (a) controlling a single main intersection, (b) coordinated control of multiple intersections along an arterial corridor, and

---

[2]Control tasks in RESCO provide the set of valid (non-conflicting) phase assignments per intersection, each affiliated with a unique index. Note that this set might differ between intersections based on their layout, unique features, and safety considerations.

[3]Signal induced delays can only be computed in hindsight. RESCO follows previous work [2] which suggested real-time approximation as the difference between the vehicle's current speed and the maximum speed limit over all vehicles.

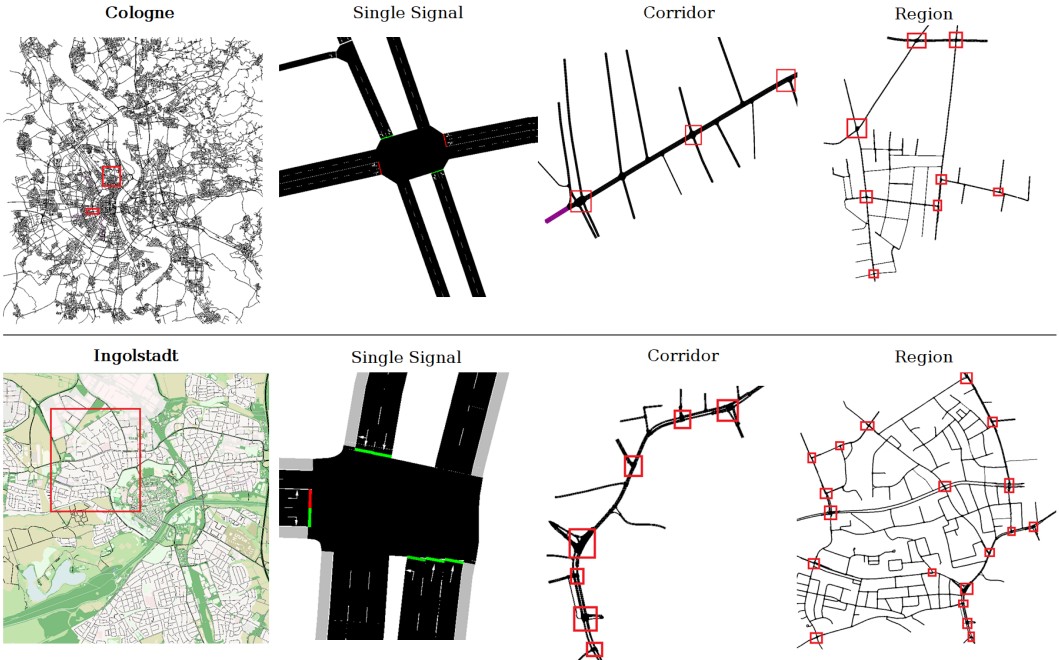

Figure 2: Road networks extracted from the full-city SUMO scenarios for benchmarking. In the full-city maps the areas bounded in red are the locations of the extracted networks. In the corridor and region networks the bounded areas mark signalized intersections. All other intersections in the extracted areas are controlled by traffic priority laws. Traffic demands for the extracted areas were taken from routes in the full-city scenarios using the standard SUMO tool for creating sub-scenarios.

(c) coordinated control of multiple intersections within a congested area (downtown). The affiliated intersections are depicted in Figure 2.

## 3.4 Benchmark algorithms

RESCO defines three baseline controllers and several RL-based controllers.

**Baseline controllers**:

(1) **Fixed-time** (or Pre-timed) control where each phase combination is enabled for a fixed duration following a fixed cycle. The intervals are defined per intersection as part of the SUMO traffic scenario;
(2) **Max-pressure** control where the phase combination with the maximal joint pressure is enabled as described in Chen et al. [7];
(3) **Greedy** control where the phase combination with the maximal joint queue length and approaching vehicle count is enabled as described in Ma and Wu [18].

**RL controllers**:

(1) **IDQN** – independent DQN agents, one per intersection, each with convolution-layers for lane aggregation as described by Ault et al. 2020. Hyper-parameters are left as the default values in the Preferred RL library with the exception of the target network update interval, which was adjusted to the Atari environment settings of 500 steps per update;
(2) **IPPO** – the same deep neural network is used as IDQN with the exception of the output layer, coming from [3]. Hyper-parameters were set following the Preferred RL defaults for the Atari environments;
(3) **MPLight** – implementation is based on the FRAP open source implementation [32] along with the ChainerRL DQN implementation and pressure sensing. Hyper-parameters are identical to that of IDQN;
(4) **Extended MPLight** – Denoted MPLight*, similar to the MPLight implementation with the addition of sensing information matching IDQN appended to the existing pressure state;

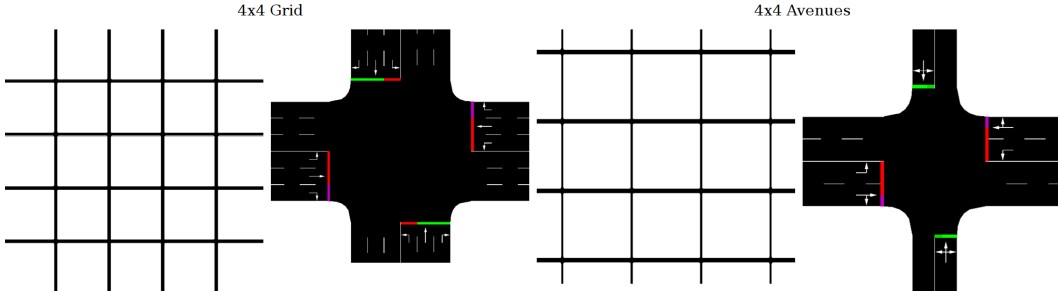

Figure 3: A $4 \times 4$ Grid validation scenario based on that presented by Chen et al. 2020 (left), and a $4 \times 4$ Avenue scenario based on the scenario presented by Ma and Wu 2020 (right).

(5) **FMA2C** – built on top of MA2C open source implementation [8]. Hyperparameters were set according to the open source implementation by Chu et al..

In each of IDQN, IPPO, and MPLight the learning algorithm implementation is called directly from the ChainerRL [11] python library successor, Preferred RL.

## 4 Experiments

The experimental section is divided into three. First, we present results that validate our benchmark algorithms implementation against performance trends reported in previous publications. Second, we provide a comparative study between state-of-the-art approaches on the realistic traffic scenarios. Finally, we draw general conclusions that characterize the algorithms and their performance.

### 4.1 Validation

For validating the benchmark algorithms implementation, we compare the learning curves and final performance of the RL controllers from Section 3.4 against the baseline controllers. The traffic scenarios are chosen to be similar to those presented in previous publications including the sensing assumptions which might change between the algorithms. Consequently, the affiliated results cannot be used for a comparative study between the RL controllers.

For validating the MPLight implementation, a traffic scenario was configured to be similar to the synthetic $4 \times 4$ symmetric network that was presented in the original MPLight publication [7]. The RESCO network is depicted in Figure 3, $4 \times 4$ Grid, and can be compared with the original network (Figure 5 in Chen et al. [7]). Demand was set according to Config. 4 in Chen et al. [7]. Figure 4 presents the learning curves for MPLight and IDQN. It also includes the baseline controllers performance (dotted lines) and the final performance for FMA2C and IPPO. FMA2C and IPPO require much more training episodes to converge (about 1,400 as opposed to 100 by IDQN and MPLight). As a result, only the final performance is included for them. The full training curves for FMA2C and IPPO are available in the appendix. Chen et al. 2020 reported a 13% improvement for MPLight over Max-pressure. The RESCO results show a similar trend with an 11% improvement.

For validating the FMA2C implementation, a traffic scenario was configured to be similar to the synthetic $4 \times 4$ traffic grid presented in the original FMA2C publication [18]. This grid is formed by two-lane arterial streets and one-lane avenues. The RESCO induced intersections are depicted in Figure 3 and can be compared with the original network (Figure 3(a) in Ma and Wu [18]). Demand was set according to the original scenario definition. Figure 4 presents the learning curve for the same set of controllers. Again, The full training curves for FMA2C and IPPO are available in the appendix.

On $4 \times 4$ Avenues validation scenario Ma and Wu 2020 reported a 4% improvement for FMA2C over Greedy and 19% improvement over IDQN. The RESCO results show a more pronounced trend for improvement over the Greedy control with a 20% improvement and a 44% improvement over IDQN. In general, both the MPLight and FMA2C implementations in RESCO present trends that follow the originally published results for each.

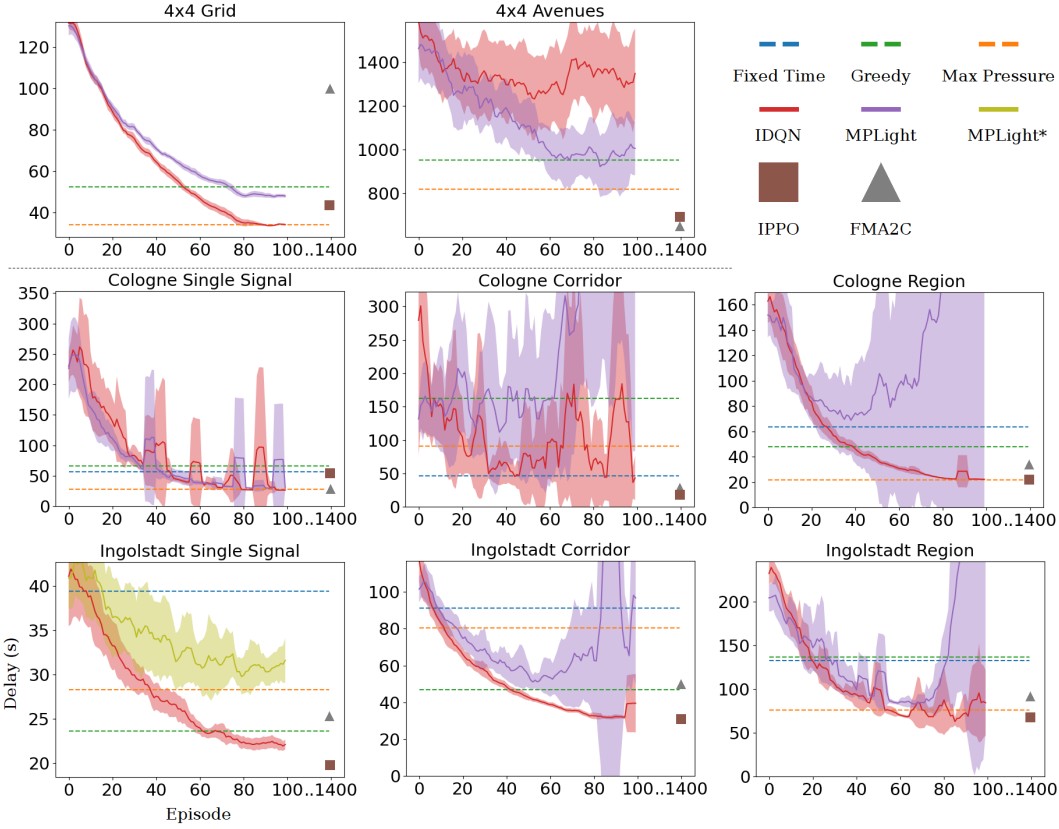

Figure 4: Learning curves over 5 random seeds, with a sliding window average of 5 episodes. Error margins display one-standard deviation from the mean delay in each episode. Baseline algorithms are marked with dashed lines, while solid lines indicate RL algorithms. Square and triangle mark the best performance of their associated algorithms, which require many times the number of episodes of the others. MPLight* denotes the extended state version of MPLight.

## 4.2 Comparative study

For the full comparative study all of the benchmark RL controllers were compared on each of the benchmark control tasks. Hyper-parameters are constant throughout and not tuned to each scenario. Following Ault et al., All sensing capabilities are assumed to be available within a 200 meter radius. The state and reward choices made by the algorithms are the primary ways in which coordination is proposed. Significantly adjusting the state or reward would therefore no longer be representative of the algorithms. Therefore the state and reward functions were set according to the definitions of each algorithm (see Section 2.3). Expanding the state to include more measurements generally decreased performance on coordination tasks. Future work should fully explore how best to augment relevant algorithms when assuming increased sensing capabilities.

Figure 4 presents the learning curves for each algorithm in every scenario while Table 1 reports the best performing episode (not necessarily the final performance) of each algorithm averaged over five random seeds. For example, consider the "Ingolstadt Region" task, Figure 4 paints a picture where MPLight diverges during training and eventually results in an average delay that is $> 200$ seconds. Table 1, by contrast, presents a delay of 78 seconds for the same task. The discrepancy is because Table 1 corresponds to the best episode, i.e., training episode 64 (before the training divergence). Table 1 also includes commonly reported metrics of delay, trip time (duration), waiting time, and queue length. However, we observe that in all but one case any one metric is sufficient to indicate superior performance in the others. Namely, in the "Ingolstadt Corridor" scenario IPPO reports improved performance over IDQN in delay, trip time, and waiting time while IDQN improves in queue length, but not to a statistically significant degree. As a result, learning curves are presented only for the delay metric (in Figure 4).

Table 1: Performance on benchmark scenarios

| IDQN | Ing. Single | Ing. Corr. | Ing. Reg. | Col. Single | Col. Corr. | Col. Reg. |
|---|---|---|---|---|---|---|
| Avg. Delay | 21.48 | 31.19 | **59.64** | **26.05** | 23.99 | 22.06 |
| Avg. Trip Time | 35.29 | 68.69 | **197.23** | **43.59** | 59.0 | 86.02 |
| Avg. Wait | 3.93 | 8.71 | **20.19** | **7.98** | 8.5 | 5.46 |
| Avg. Queue | 0.43 | **0.67** | **0.8** | **2.09** | 0.87 | 0.38 |
| **IPPO** | Ing. Single | Ing. Corr. | Ing. Reg. | Col. Single | Col. Corr. | Col. Reg. |
| Avg. Delay | **19.85** | **30.7** | 67.65 | 55.07 | **22.13** | **21.49** |
| Avg. Trip Time | **34.19** | **68.34** | 205.44 | 67.7 | **57.45** | **85.54** |
| Avg. Wait | **3.21** | **8.2** | 26.45 | 26.15 | **7.37** | **5.01** |
| Avg. Queue | **0.39** | 0.71 | 1.15 | 8.88 | **0.76** | **0.35** |
| **MPLight** | Ing. Single | Ing. Corr. | Ing. Reg. | Col. Single | Col. Corr. | Col. Reg. |
| Avg. Delay | *28.31 | 48.21 | 78.16 | 28.74 | 83.65 | 60.42 |
| Avg. Trip Time | *41.07 | 76.58 | 215.72 | 45.85 | 102.3 | 123.93 |
| Avg. Wait | *8.27 | 15.05 | 34.57 | 8.61 | 46.25 | 30.34 |
| Avg. Queue | *0.61 | 1.34 | 1.48 | 2.45 | 5.4 | 2.33 |
| **FMA2C** | Ing. Single | Ing. Corr. | Ing. Reg. | Col. Single | Col. Corr. | Col. Reg. |
| Avg. Delay | 25.36 | 48.99 | 90.42 | 30.12 | 25.37 | 33.28 |
| Avg. Trip Time | 39.4 | 85.03 | 226.5 | 47.31 | 61.68 | 97.53 |
| Avg. Wait | 7.27 | 21.9 | 44.16 | 11.23 | 11.3 | 14.19 |
| Avg. Queue | 0.94 | 1.79 | 1.74 | 3.11 | 1.68 | 0.98 |

Both Table 1 and Figure 4 report either extended state MPLight or the standard MPLight, the better performing of the two. Extended state MPLight is denoted with an asterisk. In most cases the added sensing information in extended state MPLight is not beneficial, however in the "Ingolstadt Single Signal" scenario the additional state information is beneficial and allows for convergence where standard MPLight failed to converge. MPLight, in our experiments, works well in scenarios with similarly structured intersections. However, in scenarios with varying irregular intersections we observed divergence in learning. We suspect that this phenomena is due to shared control parameters between significantly different control tasks (one controller per intersection).

IDQN achieves the best performance in the "Ingolstadt Region" scenario and "Cologne Single Signal" scenario, while IPPO achieves the best performance in all other scenarios. However, this is misleading as IPPO demonstrates significant instability. Failing to converge in the "Ingolstadt Region" and "Cologne Corridor" scenarios, and converging to significantly worse solutions in the "Cologne Single Signal". When examining the performance of each algorithm in the final 10 episodes of training, IDQN outperforms all other algorithms in all tasks except the "Cologne Regional", which IPPO still performs well in. Both FMA2C and IPPO have drastically worse sample efficiency than the DQN-based methods of MPLight and IDQN. MPLight reaches its best performing episode in 30-80% of the time IDQN does, but the performance reached is generally worse. The full training results are provided in the appendix.

## 4.3 Conclusions

Synthetic scenarios, even if made to be challenging coordination tasks, relate poorly to more realistic scenarios from the perspective of gauging the performance of reinforcement learning algorithms. Irregular and sparsely distributed signals require more general solutions than previously proposed. Deep RL methods should give increased attention to hyper-parameter sensitivity as failing to do so may result in unstable or poor performance when deployed to previously unseen scenarios.

Our experiments suggest that decentralized control algorithms are more robust compared to coordinated control algorithms when considering realistic traffic scenarios. We speculate that this is, in-part, due to the efficient data aggregation of the independent learners utilizing convolutional layers as described by Ault et al. 2020. Future work should examine merging this efficient data representation with state-of-the-art coordinated control approaches.

Coordinated approaches (FMA2C, MPLight) work well under the specific sensing assumptions that they made, but have limited applicability when advanced sensing capabilities are considered. The decentralized algorithms, IDQN and IPPO, do not share this limitation as they can effectively learn instance dependent features.

The time elapsed before an algorithm reaches its best performance is important if the ultimate goal is real-world implementation. To this end, the MPLight algorithm of Chen et al. 2020 has a considerable advantage, presenting a trade-off of reliability and converged performance for learning speed. Ma and Wu 2020's FMA2C may have an advantage when presented with challenging synthetic coordination problems, but requires a large amount of time to achieve the feat and it is questionable if scenarios fashioned from real networks and traffic demands would resemble the studied synthetic scenarios.

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
