# OpenReview forum: "Reinforcement Learning Benchmarks for Traffic Signal Control"
_NeurIPS.cc/2021/Track/Datasets_and_Benchmarks/Round1 — NeurIPS 2021 Datasets and Benchmarks Track (Round 1)_

### Official Review · Reviewer_CPDj · 2021-07-03
**Well positioned paper consolidating a diverging field**

**Rating:** 8
**Confidence:** 4
**Clarity:** Easy to follow

**Strengths:**

- Current progress in congestion control algorithms are evaluated using different settings avoiding fair comparisons between them. This is a well motivated paper that positions itself as an effort to consolidate the diverging field
- Presents a set of baselines and evaluations to outline a fair comparison between various state of art algorithms.
- Baselines are validated before they are used for comparisons.


**Weaknesses:**

- line 72-32: Be precise and revise the premise and claim about model-based methods and deep q learning algorithms
- Provide more explanation wrt yellow light -- (a) is the duration of yellow light subsumed inside a phase? (b) How is wait time etc. affected by the yellow light (c) is yellow light mandatory after every phase transitions? (d) How are "free right turn on red" etc situations handelled?
- Line 166: Missing reference and typo
- While I'm convinced, please justify why allowing algorithms the choice of state and rewards is a good choice for fair comparison give they are being evaluated on two different MDPs
- Sec 4.1 - the pronounced trend for FMA2C and IDQN isn't well justified. This is too large of a gap too gloss over


**Additional Feedback:**

- Comparative study and analysis of various methods is a bit underplayed. I'd suggest making this a prominent part of the submission. Extract it out from conclusions into its own section.

**Correctness:**

Authors evaluate the correctness of their baselines before using them for evaluations.

**Documentation:**

Open source codebase is provided


**Ethics:**

Details of licenses for prior works are provided

**Relation To Prior Work:**

Clearly outlines prior methods and explains the changes made.

**Summary And Contributions:**

- Proposes benchmarks to study the problem of congestion control using realistic traffic situations
- Presents baselines and  thorough comparative evaluations

---

> ### Author Response · Authors · 2021-07-13
> **Reply to Reviewer 4**
>
> We appreciate the time you have taken to assess our work and are encouraged that you've also recognized the need for consolidation in the area.
>
> Line 72 has been revised such that it no longer incorrectly states that DQN is an example of a model-based algorithm.
>
> More detail on yellow light functioning has been added to section 3.1. Regarding your specific questions, (a) That’s correct (b) Waiting time is increased by the duration of yellow lights (c) No, yellow lights are only inserted when required between conflicting phases (d) SUMO includes a light status to allow right-turns at red lights natively.
>
> The state and reward choices made by the algorithms are the primary ways in which coordination between signals is proposed. MPLight using pressure in the state and reward to handle all coordination and FMA2C using regional traffic liquidity for manager level rewards. Significantly adjusting the state or reward would therefore no longer be representative of the algorithms. Expanding the state to include more measurements generally decreased performance except in the case of MPLight on the single signal of Ingolstadt, which requires no coordination. Handling additional inputs could simply require larger networks, however a full exploration of augmenting the algorithms is outside the scope of this paper. Relevant text has been included in the revision.
>
> We chose to omit details of the pronounced trend for FMA2C and IDQN as the cause is presumed to be from SUMO settings which can't be confirmed for FMA2C directly. From the code available for MA2C, the base of FMA2C, it appears that there were a number of issues with the original SUMO-based implementation which makes it difficult to directly compare the performances. Specifically, in the original implementation: (a) Vehicles were allowed to teleport after waiting for a maximum length of time (b) The delay of vehicles which could not be inserted into the simulation due to congestion were not accounted for (c) The delay of vehicles which had not exited the simulation by the end of the simulation were not accounted for.

---

### Official Review · Reviewer_sWMX · 2021-07-04
**Review for Reinforcement Learning Benchmarks for Traffic Signal Control**

**Rating:** 7
**Confidence:** 3
**Correctness:** Yes
**Clarity:** The paper is clearly written.

**Strengths:**

- A standarized benchmarking on RL for traffic signal could serve as the foundation to help better future algorithm design for such problems, which could bring lots of benefits for real-world traffic control.
- The paper is well written, and the benchmarking tasks are carefully chosen to match the real-world scenarios;
- The experiments are well-conducted. A comprehensive set of baseline controllers and RL controllers are evaluated. The implementation are also validated by check against previous work, and detailed analysis are given for comparing the performance of the benchmarked algorithms.

**Weaknesses:**

- It would be interesting to see how the RL algorithms would work under weaker sensing abilities;
- Can the author give any discussion on how much sensing information mentioned in the paper can be accurately measured in the real-world traffic control scenarios. i.e., how realistic are the sensing assumptions are?



**Additional Feedback:**

no

**Documentation:**

The benchmarking is documented clearly.

**Relation To Prior Work:**

The paper gives a very detailed discussion on how it differs from previous contributions.

**Summary And Contributions:**

This paper proposes 1) benchmarking signal control tasks based on well-established traffic scenarios; 2) implementation of various RL algorithms on these signal control problems; 3) comparision and analysis of these RL algorithms under varying sensing assumptions. The benchmarking proposed in this paper could be very helpful for future study on using RL for traffic signal control.

---

> ### Author Response · Authors · 2021-07-13
> **Reply to Reviewer 3**
>
> Thank you for providing remarks, we're glad to see the agreement on the need for this benchmark. We also believe that it is important to examine the RL algorithms under reduced sensing assumptions and that will be a focus of future work. All of the measurements available in RESCO can realistically be obtained by advanced sensors available today, for more details see [13]. The installation of such sensors may not be widespread however. For that reason, RESCO enables a variety of sensing assumptions. We note that a discussion of sensing technology is outside the scope of this paper.

---

### Official Review · Reviewer_PCWe · 2021-07-06
**Review of RESCO benchmark**

**Rating:** 7
**Confidence:** 2

**Strengths:**

[S1] Paper makes a good case for the importance of studying the traffic control problem and having accurate simulators. Arguments in favour of this new simulator are clearly laid out (although note first weakness below).

[S2] On execution: Writing was generally clear, choice of experiments seemed reasonable.

**Weaknesses:**

[W1] It is unclear to me how much value this new benchmark provides for the community relative to the various existing benchmarks. The strongest argument seems to be in Section 2.3.1, which claims that existing benchmarks are less realistic because they either use simplified or arbitrarily over-complicated versions of real-world traffic grids. I do not know how much this matters for evaluation, though—it may be that the simplifications in question don't affect relative rankings of different methods at all, and the experiments in this paper do not try to evaluate how rankings change when moving from, e.g., the Jinming & Feng simulator to RESCO.

[W2] There are a few important experimental details missing from the paper.

**Additional Feedback:**

The basic idea of this paper was well-motivated and seemed well-executed. My main concern was the missing experiment details listed in the clarity & correctness sections. Traffic control is out-of-area for me, so for now I'm only comfortable giving a borderline recommendation, with low confidence.

------

(review last checked/updated 2021-07-19)

**Clarity:**

[CL1] Overall writing quality was good. Section 2.2 was helpful for understanding domain-specific nomenclature.

[CL2] It would be ideal if standard deviations were included in Table 1 (e.g. write μ±σ in each cell, or similar for radii of 95th percentile confidence intervals).

[CL3] It's not totally clear what quantities are being plotted in Table 4 just from the description. e.g. I read the top left cell as "average standard deviation of average delay". I think I understand the intention, but it's a bit confusing.

**Correctness:**

[CO1] Ideally the paper should list all the hyperparameters for all the algorithms (step sizes, architectures, batch sizes, replay buffer sizes, preprocessing, etc. etc.). Right now Section 3.4 says a lot of things like "we left all the values at defaults in the reference implementation"—this information should be included in the paper as well, ideally in a big table somewhere in the appendix.

[CO2] Relatedly, Sections 3.1 and 3.2 note that there are various choices for the observation space and reward function that the user can select between. I might have missed it, but I didn't see an explanation of which options for sensing and the reward function were selected for the experiments in Section 4. I'm also not sure how much simulated time each state transition represents (e.g. is it running at 100Hz? 0.1Hz? Variable?).

**Documentation:**

[D1] Github README was reasonably informative on options for running the experiment scripts. It would be helpful to have some Python code examples showing how to actually use the Gym environments in RESCO. Use of Markdown formatting for lists, code, etc. would also make it more readable.

**Ethics:**

I did not have any ethical concerns with the paper.

**Relation To Prior Work:**

[R1] In Section 2.3.1, the paper argues that various existing benchmarks are unjustified in claiming to be representative of "real-world city layout and demand" or having "relevancy to the real world" because they require "deeper analysis of traffic trends". I'm not sure what this deeper analysis would entail, but I didn't see anything like that in Section 3. I can see how RESCO might have an advantage over synthetic benchmarks given that it's based on simulations of real cities, but I'm not sure why it can claim to have greater realism or practical relevancy than the New York-based CityFlow environment.

**Summary And Contributions:**

This paper introduces a benchmark for RL-based control of traffic lights. The primary advantages touted over existing work include: use of the SUMO simulator; traffic scenarios based on real-world road layouts and traffic levels; a standard Gym interface; and a set of reference algorithms for the proposed environments. Experiments show that independent PPO/DQN tend to do quite well at the end of training, but take far longer to converge than specialised algorithms like MPLight.

---

> ### Author Response · Authors · 2021-07-13
> **Reply to Reviewer 2**
>
> Thank you for your thorough review. Your comments and suggestions have certainly been insightful.
>
> [W1] The main advantage of RESCO over previous environments is the ability to plug in well-established SUMO scenarios. As stated in Section 4.3, the performance trends reported by RESCO differ from those reported in previous publications for scenarios that are based on “real-world” traffic.
>
> [CO1] As suggested, we have included a table specifying all hyper-parameters in the revision.
>
> [CO2] The options used for each algorithm’s state and reward are defined by the algorithms in section 2.3. Validation experiments use sensing radii described in section 2.2. The radius for the comparative study, 200m, is the same for all algorithms as described at the beginning of section 4.2. We have restructured where this information is in the paper so it is more clear.
> The simulator can be run at variable frequencies, but here 1Hz is used. The transitions however are per action, which are in 10 second intervals. These are the same settings used by Jinming & Feng. This information has been added to section 3.1.
> Previous works have used a variety of sensing assumptions which further necessitates the need for a standard benchmark env.
>
> [CL2] Formatting and space requirements restrict this inclusion in the main text. This information, however, is now included in the appendix as Table 5.
>
> [CL3] The description has been updated in the Appendix.
>
> [R1] The referenced text was referring to the AIM simulator scenarios only. Those scenarios are perfect-grids, which might work as an approximation of a scenario for a city similar to New York. Whether they are good approximations requires further analysis and comparison against real-world trends. The New York City scenario in CityFlow uses an arbitrary selection of traffic demand amounting to roughly 25,000 vehicles per hour. This choice does not follow reports from NYC’s department of transportation indicating that approximately 30,000 vehicles enter the central business district of the city hourly (~20% discrepancy). Moreover, the central business district is only a subset of the area modeled in the CityFlow scenario, so the discrepancy is actually even larger. For this reason we believe that the realism of the scenario requires further support. The Ingolstadt SUMO scenario uses traffic data recorded by the city's government and the Cologne SUMO scenario models demands after 30,700 daily activity reports collected by the German federal government on Cologne (see [26,17] for exact scenario details).
>
> [D1] We agree. Scripts for reproducing the reported results will be included.

---

> > ### Comment · Reviewer_PCWe · 2021-07-20
> > **Response**
> >
> > Thank you for adding more results (in Table 5) and experimental details (Table 6, simulator tick rate, etc.), as well as clarifying the advantages of RESCO relative to previous work. I've upped my score accordingly, but still have low confidence because I'm not sure about the implications of using SUMO vs. other things.

---

### Official Review · Reviewer_79pK · 2021-07-06
**A potentially useful set of RL tools for Traffic Signal Control, but needs more permissive licensing and more technical work.**

**Rating:** 6
**Confidence:** 3
**Correctness:** The benchmark appears to have appropr…
**Clarity:** The paper is reasonably clear.

**Strengths:**

This system seems potentially very useful for making Traffic Signal Control easier to experiment with in an RL setting, since the OpenAI Gym API is ubiquitous, and this system handles featurization of the action and observation spaces in a convenient way.
It is also helpful that there are selected traffic scenarios covering a range of relevant realistic conditions that any algorithm attempting to solve this problem should handle.

The paper also provides a reasonable introduction to the topic, as well as a description of how Traffic Signal Control is modeled as an MDP.

**Weaknesses:**

The primary weakness of this paper is that its main contribution, the benchmark, is provided under a no-derivatives license.
This makes it illegal for other researchers to build on this work by modifying the provided code, which drastically decreases the value of the contribution.
I recognize that the individual traffic scenarios are licensed under restrictive licenses (for understandable reasons), but there's no clear reason why this restriction is present for the new contributions made in this paper.

The packaging and documentation would also both benefit from additional work.
The existing documentation consists of this paper, as well as a small readme describing some of the installation requirements and how to run the code, with the expectation that the user works out of the code directory provided.
Working out of another author's code directory is not convenient compared to being able to install the library through python's standard packaging system.
The vast majority of frequently used RL benchmarks provide the ability to install them, and it is not difficult to add support for doing so.

It would also be preferable if the different algorithms presented used a single neural network library, instead of requiring multiple ones with unclear version requirements to be installed. In particular, dependencies on versions of tensorflow that lack officially supported builds are highly inconvenient.

The paper has a few minor formatting issues (mostly unresolved references), and spends too long criticizing CityFlow, but this is not consequential.

**Additional Feedback:**

None of my objections to this paper should be difficult for the authors to fix, and I look forward to them being resolved.

**Documentation:**

The data collection and use appears ethical, since the data lacks personal information and is likely to provide public benefit. However, the hosting, licensing, and maintenance plan are all significantly lacking.
I would recommend the authors look at what high-quality RL benchmarks provide and attempt to emulate them.

**Ethics:**

I do not think there are any significant ethical concerns that warrant further discussion or review.

**Relation To Prior Work:**

The relationship to prior works is fairly clear, although I do not have sufficient knowledge to know if any significant prior work has been missed.

**Summary And Contributions:**

This contribution consists of an OpenAI Gym style wrapper around the SUMO traffic simulation package, a selection of traffic scenarios that can be run using that package, as well as a set of (RL) algorithms that can use this interface.
The authors ran all of the provided algorithms against all provided environments, and report performance figures for those runs, and make the code for doing so available.

---

> ### Author Response · Authors · 2021-07-13
> **Reply to Reviewer 1**
>
> Thank you for your thoughtful review. We agree that academic restrictive licensing is an issue. This was an honest mistake and we have accordingly updated the license to use CC BY-NC-SA, which, to our understanding, does not restrict academic usage.
>
> Easing installation via standard packaging is an excellent suggestion which we will make available as soon as possible.
>
> We understand and agree that it is inconvenient to rely on multiple NN libraries. However, we feel that there is a tradeoff between unifying the libraries and following the original implementations. It is our belief that adhering as precisely as possible to the original implementation is of higher importance.
>
> We appreciate the concern in regard to hosting, licensing and maintenance plans, we would ask if the reviewer could provide more specific details for improvement. To the best of our knowledge Github is commonly used to host benchmark environments, e.g. Arcade Learning Environment, The RoboCup Soccer Simulator, Procgen, and PyBullet Robotics Environments. If the concern is due to the repository ownership being a personal account, we do plan to move to an organizational account for the final version of the paper. The PiStar AI and optimization lab at Texas A&M will be providing long-term maintenance and support for the repository.

---

> > ### Comment · Reviewer_PCWe · 2021-07-19
> > **License**
> >
> > Thank you for changing the license. I didn't notice this on my first run through (I'm Reviewer PCWe). I don't think your license choice will pose much of a problem for academic use, but you may be interested in reading this [this stackexchange post](https://opensource.stackexchange.com/a/1718) on why CC licenses aren't more popular for code in general.
> >
> > I'd also like to +1 the suggestion that you improve the packaging. The biggest quality-of-life improvement would be to add a `setup.py` file and upload the package to PyPI so that users can install it more easily (e.g. with `pip install resco-benchmark` or similar). See [this comprehensive packaging guide for details on how to do this](https://packaging.python.org/guides/distributing-packages-using-setuptools/). I don't think Reviewer 79pK is criticising your use of Github—Github is great for sharing source code and letting others contribute, but having to manually download entire repositories and add them to your `PYTHONPATH` is a pain for end users who just want to run the benchmark. Most benchmarks I'm aware of keep their source code on Github, then periodically upload fresh releases to PyPI.

---

### Decision · Program_Chairs · 2021-07-26

**Decision:**

Accept

**Comment:**

All the reviewers appreciate the value of the proposed benchmarks. The remaining concerns seem addressable. I recommend accepting the paper while asking the authors to incorporate the review feedback into the camera-ready paper.